

# Primary molt in Gruiforms and simpler molt summary tables

Sievert Rohwer[1] and Vanya G. Rohwer[2]

[1] Burke Museum of Natural History and Culture and Department of Biology, University of Washington, Seattle, WA, United States of America
[2] Cornell University Museum of Vertebrates, Ithaca, NY, United States of America

## ABSTRACT

Molt summary tables reveal the sequence and mode of flight-feather replacement and how these feathers are divided into independent replacement series. Tables for summarizing molt are relatively new, and the rules for generating them were first formally illustrated using data from a temperate passerine. However, this first illustration failed to address (i) species with primaries divided into more than one replacement series, (ii) species with stepwise primary replacement, which almost always involves incomplete annual replacement of the primaries, and (iii) species with incomplete annual replacement within molt series characterized by single-wave replacement. Here, we review complications that arise in developing molt summary tables for such cases and we offer solutions that remove ambiguity about the direction that molt proceeds within a replacement series and about the recognition of nodal and terminal feathers that mark the beginning and end of molt series. We use these modified molt summary tables to describe the sequence of primary replacement in four groups of Gruiform birds, a group for which primary replacement has been reported to proceed from the outermost primary toward the body, unlike most other birds. Eighty molting Grey-winged Trumpeters, *Psophia crepitans,* and 124 molting Limpkins, *Aramus guarauna*, show the sequence of primary replacement is proximal in both groups; furthermore, the primaries of trumpeters are divided into two replacement series, one beginning at the outermost primary P10, and the other beginning at P3. To further evaluate the extent of this highly unusual direction of replacement in Gruiforms, we cast the data (*Stresemann & Stresemann, 1966*) on primary replacement in upland rails (Rallidae) and flufftails (Sarothruridae) into molt summary tables; both also replace their primaries proximally, from outermost to innermost, suggesting that this mode of primary replacement may be characteristic of Gruiformes.

## INTRODUCTION

Most studies of replacement patterns in the flight-feathers of birds rely on "snap-shot" data assembled across many individuals for which the age and growth of flight-feathers was recorded just once. In the past 20 years molt summary tables have revealed previously unsuspected complexities in the patterns of primary and rectrix replacement in several groups of birds. *Langston & Rohwer (1995)*, who first developed these tables, show that

Corresponding author
Vanya G. Rohwer, vgr7@cornell.edu

two North Pacific albatrosses have their primaries divided into two molt series that are replaced in opposite directions. *Shugart & Rohwer (1996)* show the primaries of night herons constitute a single replacement series, and that primary replacement in adults is stepwise, with up to four waves proceeding simultaneously through the 10 primaries. *Silveira & Marini (2012)* show the primaries of a puffbird are divided into two molt series, and two swallows and a kingbird replace their outermost rectrix, R6, as a unique series independent of the timing of molt completion for the other five rectrices (*Yuri & Rohwer, 1997*; *Imlay et al., 2017*; *Rohwer, 2008*).

These recent discoveries were facilitated by the use of molt summary tables, which efficiently summarize snap-shot observations of flight-feather replacement across many individuals in different stages of their molt. Molt summary tables show how the flight feathers are divided into replacement units (molt series), and the direction and mode of replacement within those series. Molt tables are particularly valuable because they reveal the adequacy of data used to deduce the rules of flight-feather replacement and because the rules for assembling data into these tables are sufficiently well defined to allow new data to be added to them. Molt tables have also helped clarify molt patterns in tropical species that seldom have more than a single primary growing per wave of feather renewal, and they have revealed surprising order in flight-feather replacement in species whose primary molt previously was described as chaotic, complex, random, or transilient (*Stresemann & Stresemann, 1961*; *Siegfried, 1971*; *Ginn, Glue & Hawthorn, 1974*; *Hamner, 1995*; *Herremans, 2000*). Given the importance of flight-feathers, the temporal and physiological costs of molt (*Lindström, Visser & Daan, 1993*; *Rohwer et al., 2009*), and the need to replace feathers every one, two, or sometimes three years, it is hard to imagine that selection on such an important life-history event would be haphazard.

Quantitative studies of flight-feather replacement have revealed great diversity among species (i) in how flight-feathers are organized into replacement series, (ii) in how frequently feathers are replaced in those series, and (iii) in the direction and mode of replacement within those series. Simple, distal replacement of the primaries is the rule in most groups of birds, but distal replacement is often stepwise in large birds, wherein a variable number of replacement waves proceed simultaneously through the primaries (*Dorward, 1962*; *Rasmussen, 1988*; *Shugart & Rohwer, 1996*). In other groups the primaries are divided into multiple replacement series (*Miller, 1941*; *Pyle, 2013*), and albatrosses appear to have joined the inner primaries with the outer secondaries into a single replacement series (*Edwards & Rohwer, 2005*). Despite this diversity, molt patterns of many species remain un-described, leaving the potential that molt strategies are more diverse than currently thought. For example, trumpeters, Limpkins and rails are suggested to have switched the directionality of the primary molt, wherein molt begins at the outer most primary and proceeds toward the body (*Verheyen, 1957*; *Stresemann & Stresemann, 1966*), a strategy reported for only a single other species, the Spotted Flycatcher, *Muscicapa striata* (*Williamson, 1972*).

Feather replacement strategies are likely nuanced across species, especially in non-passerine lineages, suggesting a clear role for molt summary tables to help clarify and describe these patterns. However, creating molt tables can be intimidating because, at first glance, generating these tables appears complex. With an aim of simplifying

the procedures proposed by *Rohwer (2008)* for determining the rules of flight-feather replacement using molt tables, we divide this paper into two sections. First, we provide recommendations to improve methods for casting raw molt scores into summary tables. Second, we provide the first detailed examination of primary replacement in four groups of Gruiformes—trumpeters, Limpkins, forest rails, and flufftails—that maintain the ability to fly while molting, and show that, unlike most birds, members of these groups replace their primaries proximally. Improvements in the generation of molt summary tables are well illustrated with our data on trumpeters and the clear patterns in molt strategies we present on flight feather replacement in Gruiforms.

## RESULTS

### A sequential approach to documenting molt series and replacement direction

Molt summary tables are valuable because they reveal the sample of growing feathers available at every feather locus for inferring the rules of feather replacement. For this reason, the first generation of these tables emphasized counts of growing feathers (*Langston & Rohwer, 1995*; *Shugart & Rohwer, 1996*; *Yuri & Rohwer, 1997*; *Filardi & Rohwer, 2001*). In these early tables, the direction of replacement for a growing feather was scored by the state of its neighboring feathers. This generated ambiguous scores for direction for all interior nodal and terminal feathers. By definition, nodal feathers have both neighbors old, or if neighboring feathers are growing, both are shorter than the nodal feather. In contrast, terminal feathers have both neighbors new or more fully-grown than the focal feather. Such scores from a single growing feather, flanked by neighboring feathers of the same age provide uninformative, ambiguous directionality scores because they are always associated with feathers that score as nodal or terminal. Further, scoring direction by the state of a focal feather's two neighbors is confusing because direction is intuitively understood to be inferred between adjacent feather pairs. *Rohwer (2008)* attempts to better formalize the development of molt summary tables, primarily by recognizing that directionality scores are best assessed between adjacent feather pairs, involving at least one growing feather. However, *Rohwer (2008)* advocates scoring direction of replacement at the same time that feathers are assigned nodal or terminal status. This works fine in species with complete molts but generates many contradictions in replacement direction in species with incomplete molts (e.g., *Shugart & Rohwer, 1996*) and in species with multiple waves of feather replacement (e.g., *Rohwer & Broms, 2013*). Thus, we suggest here that the first summary table should (1) summarize nodal and terminal feathers, and (2) assign direction *only* between feather pairs for which neither is nodal nor terminal. This sequence of scoring and of creating summary tables avoids the ambiguities of direction caused by transient nodal and terminal feathers, which mark points in a replacement series where molt was interrupted (see below). The advantage of this summary method is well illustrated in the molt tables we present below for trumpeters.

*Rohwer (2008)* example for generating molt summary tables fails to deal with a variety of complications that emerge when these tables are applied to species with complex or

incomplete flight-feather replacement. Thus, we begin by reviewing these complications with reference to papers that have resolved them effectively.

### Nodal feathers

Nodal feathers are highly informative because they generally mark the beginning of a molt series. However, there are stable (or dominant nodes), and transient nodes, which are distinguishable using summary tables (see below). Stable nodes mark the first feather in a replacement series. In contrast, transient nodes mark sites where molt was reinitiated following an arrest or suspension of molt. For a transient node to qualify as nodal, the arrest was probably of considerable duration because long arrests are required for the upstream neighbor to have had sufficient use to score as old. Feather age is difficult to assign in species that show little wear or foxing between cycles of active molt, like forest dwelling Gruiforms, thus transient nodes associated with short arrests will be underestimated. Feather ages may also be difficult to assign in species that replace limited numbers of flight-feathers in more than one bout of molting per year. For example, some raptors replace feathers during and after breeding (e.g., *Prevost, 1983*; *Edelstam, 1984*). Thus, replacement occurs during incubation, then molt suspends for chick feeding and resumes after chicks fledge (*Pyle, 2005*), making feather generations difficult to assign to year classes.

### Terminal feathers

Feathers that score as terminal are of little help in deducing the rules of flight-feather replacement because feathers that actually mark the end of a flight-feather series rarely score as terminal when replacement is complex. The obvious exception is the outer-most primary in passerines. Because almost all passerines replace their primaries distally from P1 to P9 or P10 and have complete molts, the outermost primary will be the last to growth and, thus will score as terminal. However, when replacement direction converges on two adjacent internal terminal feathers, only the last of the two terminals to be replaced will score as terminal; the other must be inferred to mark the end of its molt series by direction of replacement and the adjacent terminal feather in the other series (*Edwards & Rohwer, 2005* illustrate a difficult case of this for the inner two replacement series in albatrosses). When flight-feathers are broken into multiple replacement series and molt proceeds more or less synchronously and in the same direction, terminal feathers that mark the end of a series will legitimately be recognized by having newer neighbors, as is the case in cuckoos (*Rohwer & Broms, 2013*). However, such internal terminal feathers would not score as terminal if molt finishes in one series before it starts in the adjacent series. There is, as yet, no clear illustration of this problem. Despite these problems, growing feathers flanked by newer feathers should be scored as terminal because this then identifies them as feathers that should not be used in preliminary directional scoring (see next paragraph).

### Contradictory direction

When generating the first molt summary table from raw data, direction of replacement should not be scored for feathers that score as nodal or terminal. The reason is that every nodal or terminal feather with two neighbors will generate contradictory directionality scores that fail to help reveal the direction molt proceeds in a replacement series. However,

this first summary table should facilitate distinction between stable and transient nodes. Once stable nodes and terminal feathers have been identified, direction can be scored between them and their neighboring feather in the same molt series. But direction should not be scored between transient nodal or transient terminal feathers because these feathers will always generate contradictory directionality scores.

In species with multiple molt series identifying stable nodes is best done on young birds undergoing their first bout of flight feather replacement, as the location of stable nodes should be fixed and recognizable during this first molt. By contrast, transient nodes will develop in subsequent molts throughout the wing depending upon the extent of the previous molt. Stable nodes are also recognized by their high frequency compared to transient nodes associated with arrests, which, typically, are scattered throughout a replacement series (e.g., *Rohwer & Wang, 2010*; trumpeters, this paper; *Shugart & Rohwer, 1996*). It is important to realize, however, that stable nodes cannot be recognized by frequency alone when the frequency of feather replacement varies across the wing. For example, North Pacific albatrosses replace their outer three primaries annually, but they replace the two inner feathers of the outer primary series and their inner primaries and middle secondaries only every 2 or 3 years (*Langston & Rohwer, 1995*; *Edwards & Rohwer, 2005*). Thus, stable nodes can only be recognized in these feather groups by computing the probability of a feather being nodal based on how frequently it is replaced in birds that had recently completed their molt. When this was done for albatrosses, the stable nodes became readily apparent at P6, P5, and S5, with each marking the beginning of a separate replacement series (*Edwards & Rohwer, 2005*).

Stepwise molters make the generation of molt tables relatively easy because, after the first wave of replacement has reached the outermost primary, stepwise molters tend to replace all their primaries at about equal frequencies. (Stepwise replacement is largely unstudied in secondaries, but occurs in some of the secondary series in North Pacific albatrosses *Edwards & Rohwer, 2005* and other species *Pyle, 2008*). When all primaries are replaced at about equal frequency, stable nodes will be far more common than transient nodes. However, some species with multiple replacement series in the flight-feathers vary the frequency of replacement of feathers across series. When this is the case, stable nodes must be identified by rates rather than high frequencies, because rates correct for differences in the frequency that feathers in different parts of the wing are replaced (*Edwards & Rohwer, 2005*).

### Morphology may not define series boundaries

In two North Pacific albatrosses, *Edwards & Rohwer (2005)* suggest that the inner primaries and outer secondaries have been combined into a single replacement series. By contrast, for parrots and falcons, which have also divided their primaries into two replacement series, *Pyle (2013)* shows series boundaries between P1 and S1. To date, so few studies have investigated both primary and secondary replacement that we simply cannot reliably evaluate whether some feathers of these tracts are combined into a common molt series, or if molt series are defined by morphology. Scoring secondaries is extremely difficult on large birds that are prepared as traditional study skins. For this reason, we strongly encourage preserving one or both wings from large birds as fully extended separate specimens.

### Appropriate terms for molt direction

*Stresemann & Stresemann (1966)* use "ascendant" and "descendent" to describe the direction of replacement in the primaries, but these terms clearly need to be replaced by "proximal" and "distal", respectively. Ascendant and descendent continue to cause confusion in the literature, presumably because ascendant contradicts the standard numbering of the primaries, from proximal-most P1 to the distal-most primary, normally P9, P10, or P11. Thus, *Johnson & Wolfe (2018)* interpreted *Stresemann & Stresemann (1966)* to have reported distal replacement of the primaries in trumpeters, presumably relating the term to the standard numbering sequence of the primaries. Further complicating things is that some observers reverse the numbering of the primaries. Thus, *Zubergoitia, Zabala & Martinez (2018)* reported *Agolius* owls to have proximal replacement of their primaries, possibly because the reference they cite (*Korpimäki & Hakkarainen, 2012*) numbered *Agolius* primaries from outermost to innermost. Such confusion would easily be resolved by replacing "ascendant and descendent replacement" by the unambiguous terms "proximal and distal replacement".

## Primary molt in grey-winged trumpeters
### Natural history and molt scoring

Trumpeters are a family of forest birds endemic to South America and restricted to lowland regions of Amazonia and the Guiana Shield. They are rotund, chicken-sized birds with long necks and legs, and are reluctant to fly. They often live in large groups and lay 2–5 eggs in tree hollows; multiple males help raise the precocious young produced by the dominant female of the group (*Sherman & Bonan, 2018*). Trumpeters are often kept as pets or watch-dogs both because of their proclivity to give loud trumpeting calls in response to threats and because of their reputation for killing snakes. As is characteristic of many highly social animals, they are easily tamed and regularly raised from chicks (*Sherman & Bonan, 2018*). These tame birds would be ideal for further documenting patterns of flight feather replacement that cannot be scored from traditional study skins available in museums.

We assessed primary molt in trumpeters using 80 specimens from the American Museum of Natural History (AMNH). Most specimens we examined were fragile, old study skins, which limited our assessment of wing feather molt to only the primaries; we felt that examining secondary molt by scoring feather bases would have required aggressive handling of specimens, resulting in substantial damage to them.

We scored growing primaries as decimal fractions of their full length. Trumpeters have short rounded wings and seldom have adjacent primaries growing at the same time, so decimal fractions were sufficient for inferring replacement direction between adjacent pairs of growing feathers. Because trumpeters are forest birds that seldom fly, their primaries fade little between molts. Thus, we assigned feather-age mostly from wear. Feathers we judged to have been recently replaced were scored as 1 for new, while feathers judged to have been replaced in a previous episode of molting were scored as 0 for old. Birds with feathers of ambiguous age were scored as 0/1 to indicate that we were uncertain if they were new or old. In all cases where we had growing feathers to compare with full length

**Table 1  First molt summary table for the 10 primaries of 74 adult Grey-winged Trumpeters.** In this table all nodal and terminal feathers are counted, as are all directionality scores involving feather pairs that scored neither as nodal nor terminal. Note that nodal and terminal scores for P1 do not consider the status of S1, so may be in error. The columns between pairs of primaries indicate the direction of replacement (proximal: towards the body, distal: towards the tip of the wing) between feather pairs, and the number of growing feathers provides an estimate of sample size for assessing nodal, terminal, and directionality data for each feather. P10 is the outer most primary and P1 is the inner most.

| | P1 | | P2 | P3 | | P4 | | P5 | | P6 | | P7 | | P8 | | P9 | P10 |
|---|---|---|---|---|---|---|---|---|---|---|---|---|---|---|---|---|---|
| # Nodal feathers | 6 | | 3 | 9 | | 1 | | 1 | | 2 | | 0 | | 1 | | 2 | 12 |
| # Terminal feathers | 3 | | 2 | 0 | | 2 | | 3 | | 1 | | 0 | | 1 | | 1 | 3 |
| # Proximal directionality | | 0 | | | 2 | | 3 | | 7 | | 6 | | 8 | | 11 | | |
| # Distal directionality | | 0 | | | 0 | | 1 | | 0 | | 1 | | 0 | | 0 | | |
| # Growing feathers | 6 | | 4 | 7 | | 5 | | 7 | | 8 | | 3 | | 8 | | 11 | 6 |

feathers, the color and wear status of fully grown feathers was compared to that of the growing feathers to assure as much accuracy as possible in assigning feather ages.

Fully-grown birds were aged as first-years or adults on the basis of the width and shape of the rectrices. Juvenile rectrices are narrower and more pointed than basic feathers of adults (S Rohwer & V Rohwer, pers. obs., 2018). For age determination we carefully examined all the rectrices, as some first-years were in the process of replacing their tail feathers. There was no ambiguity about birds that were fully grown, and we found just a single downy chick among the 80 fully grown Grey-winged Trumpeters in the AMNH collection. Our raw data summarizes the molt status of 74 adults and six first-year individuals, and is presented as a Supplemental Information 2.

*Trumpeter molt tables*

We developed the molt summary tables from the raw data in two steps, as proposed above. First, for adults, we recorded and counted all nodal and terminal feathers in the raw data, but assigned directionality score only between feather pairs that included neither nodal nor terminal feathers (Table 1). This table clearly reveals two dominant nodes in the primaries, P10 and P3, marking the first feather in each molt series. P10 was nodal in 12 specimens and P3 was nodal in nine specimens. In contrast, feathers between P9 and P4 scored as nodal from zero to two times, suggesting that these were transient nodes associated with the re-initiation of molt following arrests. P3 scored as nodal nine times, but this frequency can only be compared to that for P2, which has two scored neighbors. P2 received three nodal scores and two terminal scores, and these low nodal scores compared to P3, suggest that P3 is the first feather of a replacement series that includes the inner primaries. The six nodal scores associated with P1 are addressed below.

The leap from Table 1 to Table 2 is short, involving just three steps. First, because directionality between all feather pairs between P9 and P4 was strongly proximal, it is appropriate to add directionality scores between P10 and P9, which are proximal in seven cases and distal in three. It might seem that the three distal scores suggest problems of interpretation, but P10 is rather short compared to P9. Thus, P10 may sometimes complete growth before P9, even when P9 is lost after P10, because P9 takes longer to grow. Second, directionality scores between molt series should be dropped (see above and *Rohwer, 2008*). This eliminates the two cases of proximal directionality between P4 and P3; in both cases

**Table 2** **Final molt summary table for the primaries of 74 adult Grey-winged Trumpeters.** In this table we have added directionality scores between nodal and adjacent feathers, removed the directionality scores between terminal P4 and nodal P3, and added a grey bar to mark the series break between P4 and P3. See Table 1 legend for additional definitions.

|  | P1 | P2 | P3 | P4 | P5 | P6 | P7 | P8 | P9 | P10 |
|---|---|---|---|---|---|---|---|---|---|---|
| # Nodal feathers | 6 | 3 | 9 | 1 | 1 | 2 | 0 | 1 | 2 | 12 |
| # Terminal feathers | 3 | 2 | 0 | 4 | 3 | 1 | 0 | 1 | 1 | 3 |
| # Proximal directionality | 0 | 9 | | 3 | 7 | 6 | 8 | 11 | 7 | |
| # Distal directionality | 0 | 0 | | 1 | 0 | 1 | 0 | 0 | 3 | |
| # Growing feathers | 6 | 4 | 7 | 5 | 7 | 8 | 3 | 8 | 11 | 6 |

P4 was growing and P3 was old. A grey bar can now be added between P4 and P3 to identify the boundary between the outer and inner replacement series in the primaries, and the two proximal scores between P4 and P3 can be inferred as terminal scores for P4, increasing the terminal counts for P4 from two to four. Third, with P4 established as terminal in the outer replacement series, it is now appropriate to assign replacement direction between P3 and P2; these scores are proximal in nine out of nine cases. Note that these directionality scores would have all been contradicted by scores of distal between P3 and P4, had they been applied before establishing that P4 was the terminal feather in the outer primary molt series. Samples of adult trumpeters molting the inner most primaries are small, so our inference of proximal directionality between P2 and P1 is weak. The directionality scores between P3 and P2 are strongly proximal, suggesting directionality between P2 and P1 should be similar; indeed, in one first-year (AMNH 125285), directionality between P2 and P1 is proximal.

There are just six first-year individuals among the 80 specimens of Grey-winged Trumpeters in the AMNH collection. One of these first-years, AMNH 125285, is initiating molt at P10 and P3, thus confirming that these feathers mark the start of the two replacement series in the primaries. AMNH 431790 failed to confirm the series we identified using adults, but we probably erroneously aged P3 through P1 as juvenile feathers, instead of new feathers. This individual was in advanced stages of molt; all outer primaries had been replaced and P4 was already halfway grown, suggesting that the inner series may have already completed molt. Recall that feather ages were difficult to assign because they show little wear or fading; thus, in the case of birds aged as first-years, we tended to record feathers that appeared to be old as j (for juvenile primary), rather than 0 (old) or 1 (new).

Why did P1 frequently receive nodal scores? In reality these nodal scores may be inappropriate if the inner primaries and the outer secondaries are part of a single replacement series beginning at P3, as suggested for North Pacific albatrosses (*Edwards & Rohwer, 2005*). Because we could not score secondary replacement in the round skins at the AMNH, we could not evaluate the timing and sequence of replacement between P1 and S1, as is needed to evaluate linkage between them. Further, S1 is nodal in so many groups of birds that speculation on this point seems fruitless without having data from birds actively replacing their outer secondaries (see *Edwards & Rohwer, 2005*; *Pyle, 2013*).

**Table 3  Number of feathers growing in the inner and outer molt series in 74 adult Grey-winged Trumpeters.** This table summarizes the number of primaries growing (0–3) in the outer (P10–P4) and inner (P3–P1) molt series. Birds that were actively molting most often grew one feather at a time.

|  | Number of feathers growing per series | | | | |
|---|---|---|---|---|---|
|  | 0 | 1 | 2 | 3 | Totals |
| P10-P4 series | 36 | 31 | 6 | 1 | 74 |
| P3-P1 series | 56 | 18 | 0 | 0 | 74 |

### Primary replacement is stepwise in the outer series

The primaries from P10 to P4 are a single molt series replaced proximally. Primaries P9 through P4 received scores of nodal or terminal from zero to three times. The regular occurrence of transient nodal and terminal feathers between P9 and P4 strongly suggests that primary replacement in trumpeter's outer replacement series is stepwise and, further, that molt is regularly interrupted and restarted, both where it was arrested and also at P10. A consequence of stepwise primary replacement is that all feathers of stepwise replacement series are eventually replaced at approximately equal frequencies. Recall that stepwise replacement eventually generates approximately equal frequencies of replacement in adults, making stable nodes recognizable by frequency alone, without having to adjust for rate differences in replacement to find stable nodes, as is necessary in albatrosses (*Edwards & Rohwer, 2005*).

From studies of a very limited number of first-year individuals from 13 species of large birds undergoing their first replacement of the primaries, we know that, in birds that feature stepwise primary replacement, multiple waves of replacement develop following molt arrests (*Rohwer, 1999*; *Pyle, 2006*). Thus, after each arrest, molt is reinitiated where it stopped in the first (or later) bout(s) of molting, and also at the first primary in the molt series. In the case of trumpeters, with its unusual proximal direction of replacement, these new waves would start at P10.

We checked for active molt in the outer and inner primary series in 74 adults. Thirty-six individuals were not growing primaries, while 38 were growing one or more primaries in the outer series (Table 3). Of the 38 growing primaries, 31 had just a single wave of replacement, six had two waves, and one had three waves of replacement, showing that molt is often reverse stepwise in trumpeter's outer primary series. In the inner primary series, which could include some secondaries, none of the 18 birds replacing feathers between P3 and P1 had more than a single wave of replacement. Note that, with just three inner primaries, the likelihood of recording two waves so close together is very small, especially with just 18 birds showing active molt in these feathers.

The case for stepwise replacement in the outer primaries is further supported by patterns of old and new feathers in that series. Of the 36 adults that were not growing primaries in the outer series, 26% had a mix of old and new feathers, demonstrating molt arrests. This percent closely matches the 23% of specimens in active molt that were replacing their primaries in two or three waves. Most of the 36 specimens that were not in active molt had single waves of newish feathers that abutted or lay between blocks of old feathers, but at least one showed evidence of molt arrests involving more than a single replacement wave.

### Production seems very low in trumpeters

There were 80 fully grown specimens of Grey-winged Trumpeters in the collection at the AMNH, all of which we aged by rectrix shape. We were surprised to find just six birds in this collection with either all juvenile rectrices ($n = 1$) or a mix of juvenile and adult rectrices ($n = 5$). Because first-years closely resemble adults in appearance and because we could only age these collected specimens with reference to rectrix shape, it seems unlikely that collectors were preferentially targeting one or the other of these age classes. Thus, production of what we assume are independent young seems to be very low in this trumpeter species, at about 0.075 first-years per adult per year. Trumpeters live in groups, with several males assisting in rearing the chicks produced by a single female, so some of the adults we examined were surely not breeders. Nonetheless, this seems like a very low rate of production, suggesting that mature birds are long-lived. Unfortunately, we do not know how long juvenile rectrices are retained in these birds; if they are replaced by adult rectrices before young are a year of age, then our estimate of production would be low. *Johnson & Wolfe (2018)* suggest that the juvenile rectrices are not replaced in their first molt, which, if true, would suggest that this estimate of production is high.

## Primary molt in limpkins
### Natural history and molt scoring

Like trumpeters, the Limpkin, Aramus *guarauna*, is an exclusively new world Gruiform species that inhabits swamps and slow-moving river systems. It is sometimes polygynous (*Bryan, 2002*) and famous for its mournful, screaming cries. On the basis of five specimens *Stresemann & Stresemann (1966)* described its molt as ascendant, or proximal, but their data were insufficient for critical analyses. Further, *Verheyen (1957)*, who was so often a pioneer in molt studies, reported the direction of their primary replacement to be distal. To definitively establish their mode of primary replacement we scored all specimens in the AMNH, the Cornell and Burke collections, and a series at the USNM, and summarized those data following procedures described above for trumpeters. In adult Limpkins outermost P10 is sickle shaped, but of little use in determining age for molt studies because it is the first primary to be replaced. Thus, we aged all of the specimens by rectrix width (*Pyle, 2008*), making it possible to distinguish birds in their first primary replacement from adults. Our raw data are presented as supplemental material.

### Limpkin molt tables

The molt summary table for adults shows the direction of primary replacement to be strongly proximal, usually commencing with the loss of P10, although P9 is lost before P10 in some birds. There were good samples of growing primaries at most loci, but fewer among the inner primaries (Table 4). P10 was strongly nodal and is clearly the point at which primary replacement usually starts. The data for adult Limpkins suggest the possibility that P8 might mark a node for an additional replacement series. P8 scored as nodal 6 times, higher than all other primaries except P10. However, there are so many interruptions of molt in adults that interpreting these numbers to suggest the primaries are broken into two replacement series could easily be erroneous. Indeed, there was no suggestion that P8 was nodal among a large sample ($n = 48$) of young birds replacing their juvenile primaries,

**Table 4  Molt summary table for 76 adult (top) and 48 immature (bottom) Limpkins.** This table summarizes nodal and terminal feathers, the direction of feather replacement (proximal or distal), and gives of sample sizes used to calculate nodal, terminal, and directionality data by showing the number of growing feathers for each primary across all adult (top) and immature (bottom) Limpkins we examined.

| | P1 | P2 | P3 | P4 | P5 | P6 | P7 | P8 | P9 | P10 |
|---|---|---|---|---|---|---|---|---|---|---|
| **Adults** | | | | | | | | | | |
| # Nodal feathers | 1 | 1 | 0 | 0 | 1 | 2 | 1 | 6 | 1 | 12 |
| # Terminal feathers | 0 | 1 | 2 | 2 | 2 | 1 | 2 | 5 | 5 | 0 |
| # Proximal directionality | 2 | 4 | 4 | 6 | 12 | 11 | 7 | 7 | 10 | |
| # Distal directionality | 0 | 0 | 0 | 0 | 0 | 0 | 0 | 0 | 0 | |
| # Growing feathers | 2 | 4 | 6 | 4 | 9 | 9 | 11 | 9 | 11 | 11 |
| **Immatures** | | | | | | | | | | |
| # Nodal feathers | 0 | 0 | 0 | 0 | 0 | 0 | 2 | 0 | 1 | 6 |
| # Terminal feathers | 0 | 0 | 0 | 0 | 0 | 0 | 0 | 2 | 0 | 0 |
| # Proximal directionality | 0 | 1 | 1 | 0 | 1 | 4 | 5 | 6 | 4 | |
| # Distal directionality | 0 | 0 | 0 | 0 | 0 | 0 | 0 | 0 | 1 | |
| # Growing feathers | 0 | 0 | 1 | 0 | 0 | 1 | 4 | 3 | 6 | 8 |

as would be expected if the primaries were broken into two replacement series (Table 4). Thus, we conclude the primaries of Limpkins are all part of a single molt series, replaced proximally from P10.

Because adult Limpkins show frequent arrests in primary replacement, it is unsurprising that stepwise replacement of the primaries is common. Of the 76 adults we scored, 38 were in active molt. Of these 38, 20 had two waves of primary molt proceeding simultaneously through a single wing, 12 had one wave, and six had three waves. Similarly, of the remaining adults not in active molt ($n = 38$), 20 had a mix of new and old feathers, which would generate two waves of primary replacement upon reinitiating molt, mirroring data for Limpkins in active molt. Actively molting Limpkins commonly replaced between one and four feathers per wing at once, and one individual was growing five feathers simultaneously on one wing.

Proximal replacement of the primaries in Limpkins resulted in more frequent replacement of the outer primaries compared to the inner primaries (Fig. 1). While adult Limpkins were collected more frequently in February and March (average of 10 adults/month), this uneven sampling probably does not bias estimates of replacement frequency of primaries, as the number of specimens are relatively evenly spaced for the remaining months (average of ∼5 adults/month). P10 is also the shortest primary, which suggests it should receive fewer scores of "growing" compared to other primaries that take longer to grow, yet it was frequently scored as growing. Clearly, proximal molts result in the frequent replacement of the outermost primaries, the feathers that receive the most wear. That P1 scored more frequently as new compared to other inner primaries may be a result of this inner feather receiving less wear and erroneously being scored as new. Alternatively, P1 could be part of an additional series with the secondaries; this alternative awaits data that includes the secondaries.

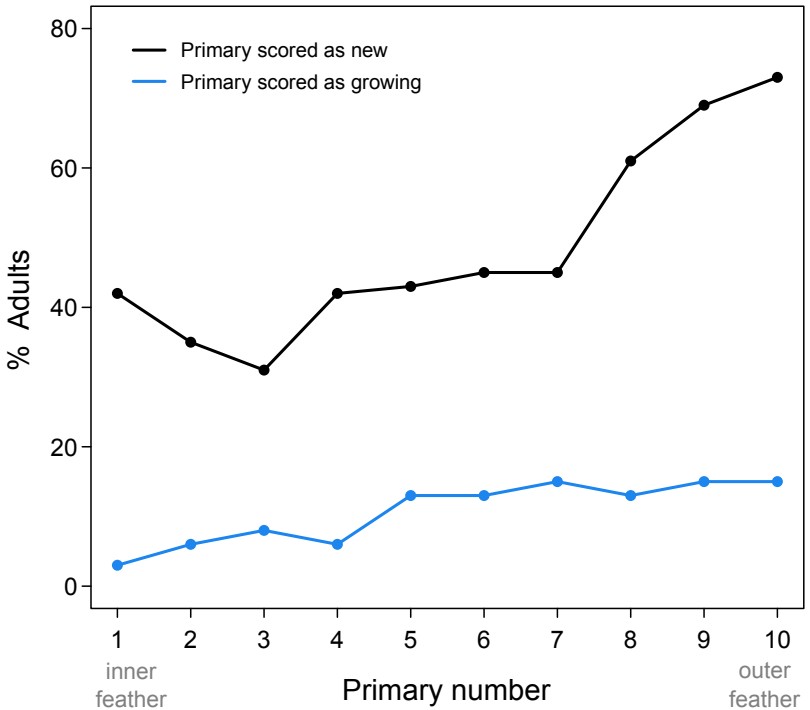

**Figure 1** **Summary of new and growing primaries from 76 adult Limpkins.** Figure plots the percentage of adults with new or growing primaries across the wing. As predicted by proximal primary replacement, outer primaries, where molt initiates, are more frequently scored as new or growing compared to inner primaries, indicating more frequent replacement of outer feathers.

## Primary molt in flufftails and in rails that fly while molting
### Natural history and molt scoring

Most rails molt their primaries more or less synchronously, as is characteristic of many water birds that can feed and escape predators during a brief period of flightlessness associated with synchronous or near-synchronous loss and regrowth of flight-feathers. *Verheyen (1953)* seems to have been the first to recognize that some forest rails retained the ability to fly while replacing their wing quills, and his observation inspired *Stresemann & Stresemann (1966)* to undertake a very thorough examination of primary replacement in upland rails and flufftails. Upland rails are found in many tropical regions of the world, while flufftails, which are now placed in their own family, are relatives that live in dense forests and thick grasslands of sub-Saharan Africa and Madagascar. As we found for trumpeters, the Stresemanns often could not determine the age of primaries in rails and flufftails. Thus, they recorded such feathers as "standing", which we denote as present in our summary of their data. When the neighbor of a growing primary was recorded as standing, directionality between the pair could not be scored and, when both of a growing feather's neighbors were scored as "standing", it could either have been nodal or terminal, so we scored it ambiguously as "n/t". Despite these difficulties, trends in the Stresemanns' data for rails and flufftails are strong when pooled across species within these families, even though samples of molting birds were small for most of the species treated.

**Table 5  Molt summary table for 52 individuals of 14 species of forest dwelling rails.** This table summarizes data presented in *Stresemann & Stresemann (1966)*. Definitions of nodal and terminal feathers, directionality scores, and the number of growing feathers (i.e., sample sizes) are the same as those presented in Table 4. Because feather ages were sometimes difficult to assess or not provided, we indicate uncertainties in nodal/terminal or directionality assignments in rows: # Nodal/terminal? and # ? directionality.

| | P1 | P2 | P3 | P4 | P5 | P6 | P7 | P8 | P9 | P10 |
|---|---|---|---|---|---|---|---|---|---|---|
| # Nodal feathers | 0 | 1 | 0 | 0 | 3 | 0 | 0 | 1 | 2 | 9 |
| # Terminal feathers | 10 | 0 | 0 | 0 | 1 | 1 | 0 | 0 | 1 | 5 |
| # Nodal/terminal? | 3 | 1 | 2 | 2 | 2 | 0 | 1 | 0 | 1 | 0 |
| # Proximal directionality | 10 | 11 | 16 | 20 | 14 | 20 | 27 | 17 | 9 | |
| # Distal directionality | 0 | 0 | 0 | 1 | 2 | 1 | 0 | 0 | 7 | |
| # ? directionality | 5 | 18 | 21 | 27 | 24 | 28 | 35 | 26 | 13 | |
| # Growing feathers | 9 | 16 | 17 | 21 | 19 | 23 | 25 | 23 | 25 | 16 |

Despite being extraordinary in scope, the *Stresemann & Stresemann (1966)* data for primary replacement in rails has largely been lost to science because they provided only verbal descriptions of primary replacement for each specimen examined; further, their molt descriptions are far from parallel across species and groups, and across museums that they visited at different times in their studies. In a strongly positive review of their classic monograph, *Stettenheim (1969)* noted that their text descriptions of each individual specimen in wing molt made the perception of patterns in the data largely impossible. Thus, in response to Peter Pyle's review of an earlier draft of this paper, we have cast all data on primary replacement in rails and flufftails presented in *Stresemann & Stresemann (1966)* into molt tables. For many specimens the Stresemanns found substantial asymmetries between wings. When both wings were scored and molt was asymmetrical, we have included both wings in the supplemental data file and in our molt summary tables. While this may seem to constitute pseudo-replication, the numbers in our summary tables make it obvious that results would be unchanged had we arbitrarily dropped one of these wings; including them in the supplemental data files makes all of the Stresemanns' data for these groups accessible to the world's ornithologists.

### Rail primary replacement

There were large samples of growing feathers at every primary locus in the Stresemanns' composite data for the rails lacking synchronous molt (Supplemental Information). Primary replacement in upland rails that retain the ability to fly while molting is very clearly proximal. For every primary pair except P9/P10, proximal directionality scores strongly predominate over distal scores (Table 5). For P9/P10 there were nine proximal scores and seven distal scores. This near equality probably suggests that P9 is sometimes dropped before P10, but it may also reflect P10 being shorter than P9 in most rails, and could be a scoring error; the Stresemanns often scored feathers as fractions of their full lengths, without presenting the relative lengths of the feathers. Clearly, however, primary replacement is initiated with the loss of P9 or P10 and proceeds proximally at least to P6.

There is weak evidence that P5 may constitute an interior node, which would suggest the primaries may be broken into two replacement series, P10 through P6 and P5 through P1 (Table 5). P5 is nodal three times, while none of P1-P8 was nodal more than once.

**Table 6  Molt summary table for eight individuals of three species of flufftails.** This table summarizes data presented in *Stresemann & Stresemann (1966)*. Nodal and terminal feathers, directionality scores, and sample sizes (i.e., the number of growing feathers) follow definitions in Table 4. We indicate uncertainties in nodal/terminal or directionality assignments in rows: # Nodal/terminal? and # ? directionality.

|  | P1 | P2 | P3 | P4 | P5 | P6 | P7 | P8 | P9 | P10 |
|---|---|---|---|---|---|---|---|---|---|---|
| # Nodal feathers | 0 | 0 | 0 | 0 | 0 | 0 | 0 | 0 | 1 | 3 |
| # Terminal feathers | 0 | 1 | 0 | 0 | 0 | 0 | 0 | 0 | 2 | 0 |
| # Nodal/terminal? | 0 | 0 | 0 | 1 | 1 | 0 | 0 | 1 | 0 | 0 |
| # Proximal directionality |  | 0 | 1 | 2 | 2 | 1 | 1 | 2 | 2 | 3 |
| # Distal directionality |  | 0 | 0 | 0 | 0 | 0 | 0 | 0 | 1 | 0 |
| # ? Directionality |  | 0 | 0 | 0 | 0 | 0 | 0 | 0 | 1 | 1 |
| # Growing feathers | 0 | 1 | 1 | 1 | 2 | 1 | 1 | 3 | 4 | 3 |

But this inference is countered by the direction of replacement between P6 and P5, which was proximal 14 times and distal just two times. If there were two molt series, then, to minimize time in molt (*Rohwer & Broms, 2013*), both series would be expected to commence at about the same time and, if that were the case P6 should usually be old when P5 is growing, suggesting distal replacement should predominate between this primary pair. But direction between P6 and P5 was overwhelmingly proximal, so we conclude that the primaries of rails are not broken into two replacement series.

The scattering of internal nodes and termini between P8 and P2, as well as the growing feathers scored as n/t (because their neighbors' ages could not be scored) reflect the fact that primary loss often occurs out of the general proximal sequence (Table 5); indeed, the Stresemanns often describe individual species as having proximal replacement with irregularities common, and in some species they note that the secondaries are lost nearly simultaneously (see Supplemental Information).

***Flufftail primary replacement***

The data available in *Stresemann & Stresemann (1966)* on primary replacement in flufftails is quite limited, amounting to eight single wings and one bird scored for both wings (see Supplemental Information). Despite scores for just nine wings, there were from one to four growing feathers for primary loci P2 through P10; no specimen was growing P1 (Table 6). Only P10 was nodal and direction of replacement was proximal in 14 cases between P2 and P10, and distal in just a single case, leaving little doubt that primary replacement in flufftails is proximal. The only case of distal direction was associated with AMNH 546274, a specimen for which replacement was nearly simultaneous (*Stresemann & Stresemann, 1966*). Although P10 was clearly the dominate node marking the initiation of molt, this composite data set has too few growing feathers among the inner primaries to evaluate the possibility of one or more interior nodes that could mark divisions of the primaries into two or more molt series.

## DISCUSSION

We accomplish two things in this paper. First, we refine earlier methods for developing molt summary tables that clearly show the data and reasoning behind inferences about

modes of flight-feather replacement. For many small temperate passerines with intense primary molts, molt tables may seem unnecessarily complicated. However, they do reveal the adequacy of the data for making inferences about molt series and replacement direction within series. For many birds with more complex patterns of flight-feather replacement, summary tables are essential for uncovering complexities that otherwise have gone unnoticed. This is well illustrated for albatrosses (*Langston & Rohwer, 1995*; *Edwards & Rohwer, 2005*), night herons (*Shugart & Rohwer, 1996*), falcons and parrots (*Pyle, 2013*), a tree swift (*Rohwer & Wang, 2010*), and cuckoos (*Rohwer & Broms, 2013*). We used these studies and others to summarize some of the caveats that need to be considered when transcribing raw molt scores into summary tables. With regard to the generation of molt summary tables, our principle contribution in this paper has been to show that direction of replacement should not be scored between transient nodes or termini and their neighbors, simply because the two directionality scores will always be contradictory.

Second, we present quantitative data showing that four Gruiform families—trumpeters, Limpkins, rails and flufftails—replace their primaries proximally, as suggested by *Stresemann & Stresemann (1966)*. The Stresemanns recognized that primary replacement was proximal in some of the families that were then placed in Gruiformes, and made a special effort to evaluate the mode of primary replacement in upland rails that retain the ability to fly while molting in case these rails replaced their primaries proximally. Our summaries of their data clearly show that primary molt is proximal in forest rails and flufftails, suggesting similar molt strategies to other core groups in modern Guiformes. Molt tables also confirm earlier suspicions that trumpeters and Limpkins replace their primaries proximally. Unfortunately, we have found no data that address either the sequence of primary replacement in cranes, or whether their primaries may be broken into multiple replacement series. Some cranes undergo simultaneous primary molts, often after two or three years of feather use (*Folk et al., 2008*; *Lewis, 1978*), and Sandhill Cranes (*Grus canadensis*) sometimes alternate between simultaneous and sequential replacement of their primaries, a pattern also observed in some flamingos (*Studer-Thrush, 2000*). Unfortunately, however, none of the molt studies for cranes reports direction of replacement or whether their primaries may be broken into more than one replacement series.

In summary, all Gruiforms that have been studied and that retain the ability to fly while molting primaries replace their primaries proximally, from P10 inward. Living Gruiformes are divided into two suborders the Grui, which includes Psophidae, Aramidae, and Gruidae, and the Rali, which includes Heliornithidae (all three genera apparently have simultaneous wing molts; *Stresemann & Stresemann, 1966*), Sarothruridae and Rallidae (*Fain, Krajewski & Houde, 2007*). Although the sequence of primary replacement remains unestablished for Gruidae, proximal replacement of the primaries seems to be an ancestral trait in the Gruiform lineage. That Grey-winged Trumpeters have their primaries divided into two molt series may be a derived condition in the suborder Grui, but this possibility cannot be evaluated without knowing how primary replacement in Gruidae that fly while molting is organized.

As the Stresemanns observed, cranes are such big and cumbersome specimens that there are essentially none in collections that are molting flight feathers. Thus, we strongly

recommend the preservation of extended wings of any crane available to be preserved as a scientific specimen. Extended wings require little time to prepare, are efficient to store, and make it possible *and* easy to score secondary molt. They also make it easy to account for the fact that the number of secondaries may vary substantially among individuals in species of large birds with long wings (e.g., albatrosses: *Edwards & Rohwer, 2005*). Extended wings also greatly facilitate measuring growth bands, which, when faint, are impossible to measure on traditional study skins and which are essential for estimating feather growth rate and the duration of flight-feather molt (e.g., *Rohwer & Broms, 2012*; *Rohwer & Broms, 2013*).

Our molt summary tables revealed that trumpeters have divided their primaries into two replacement series, the outer of which is clearly P10–P4; the inner starts at P3, but its terminal feather could not be defined as we were unable to evaluate secondaries in these specimens. The division of the primaries into two replacement series was a surprise and may be an adaptation to allow more rapid renewal of the flight-feathers, as occurs in cuckoos (*Rohwer & Broms, 2013*). Alternatively, division of the primaries into two series could be an adaptation that allows feathers in the middle of the wing to be replaced less often than feathers at the tip of the wing that suffer more wear, as occurs in some albatrosses, where middle molt series are not activated every year (*Langston & Rohwer, 1996*; *Edwards & Rohwer, 2005*).

So far as we know proximal replacement of the primaries has been reported in only one other bird, the Spotted Flycatcher, *Muscicapa striata*, of Eurasia (*Williamson, 1972*). Why should primary replacement proceed proximally in these Gruiform families and in the Spotted Flycatcher, when it proceeds distally in most other birds? For Limpkins, *Howell (2010)* has suggested proximal replacement could facilitate early replacement of P10, which is sickle-shaped in adults and may be used in display. Alternatively, Limpkins and other Gruiforms live in rather dense vegetation where flight may damage outer primaries more than the middle wing feathers. If that were the case, then selection could favor renewing the outer primaries first because proximal directionality results in frequent replacement of outer feathers that receive the most wear. Note, however, that both suggestions seem implausible because they should apply to many other groups of birds. Further, proximal replacement of the primaries seems to be a shared derived character in Gruiforms, meaning that special interpretations related to families within this order are likely to be specious. Unless more phylogenetically unrelated groups are discovered in which the outer primaries are replaced proximally, there seems little chance that proximal and distal replacement of the primaries can successfully be associated with potential selective regimes through comparative studies. Perhaps more likely is that their unusual replacement direction arose from some accidental gene inversion, which drifted to fixation because the direction of primary replacement has minimal fitness consequences.

## CONCLUSION

The detailed descriptions of molts provided by *Stresemann & Stresemann (1966)* were originally aimed to elucidate evolutionary relationships among birds. However, molt data appeared too complex and clumsy to be amenable to comparative studies; different species

had different molt strategies and subtle, but important, differences often went unnoticed with historical methods of documenting molt. Molt sequence data is now confirming evolutionary relationships among birds (*Pyle, 2013*; *Sibley & Ahlquist, 1990*), one of the original goals of the Stresemanns' efforts, and molt tables are helping to clarify subtle differences in molt strategies across species. In our treatment of four Gruiform families, all have proximal primary replacement suggesting that this molt sequence is an ancestral trait in Gruiformes. One member, the Grey-winged Trumpeter, appears to combine multiple series and stepwise molt, a strategy that to date has been recorded in only a few other groups (*Pyle, 2013*). This unexpected complexity in the pattern of primary replacement represents an important evolutionary transition for organizing the replacement of flight-feathers, yet it likely would have been overlooked without the use of molt summary tables. This finding suggests that the multiple molt series of trumpeters is a derived molt strategy in Gruiforms, but this suggestion requires examining more members of the Gruiform clade. As more such reports accumulate, the stage will be set for a comparative analysis of the transitions between modes of flight-feather replacement across living birds.

## ACKNOWLEDGEMENTS

This study is based on specimens at the American Museum of Natural History (AMNH), the US National Museum (USNM), the University of Washington Burke Museum (UWBM) and the Cornell University Museum of Vertebrates (CUMV). Thanks to Paul Sweet for facilitating our visit at the AMNH, and to Ernest Power for giving us a place to stay in New York City during our work at the AMNH. Inge Roberts was an enormous help in transcribing the Stresemanns' descriptions of primary replacement in rails and flufftails into the molt summary table we present in the online supplemental material, and Sarah Dzielski and Lea Callan were wonderfully helpful in recording Limpkin data. Peter Pyle provided a very helpful review of the original manuscript, which inspired us to include Limpkins, rails, and flufftails in this paper and to transcribe the Stresemanns' data on rails and flufftails into accessible summary tables.

### Funding
The authors received no funding for this work.

### Competing Interests
The authors declare there are no competing interests.

### Author Contributions
- Sievert Rohwer and Vanya G. Rohwer conceived and designed the experiments, performed the experiments, analyzed the data, contributed reagents/materials/analysis tools, prepared figures and/or tables, authored or reviewed drafts of the paper, approved the final draft.

## Data Availability

The raw data are provided in a Supplemental File.

## Supplemental Information

Supplemental information for this article can be found online at http://dx.doi.org/10.7717/peerj.5499#supplemental-information.

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
