# Peer review of "Primary molt in Gruiforms and simpler molt summary tables"

_PeerJ, doi:10.7717/peerj.5499_

## Round 0.1 · original submission · Minor Revisions

I'll start with a quick apology for the lateness of this decision. The reviews came back right in the middle of a family emergency. I thought I could get to your manuscript within about 3 or 4 days and then make the decision. It ended up sitting on my desk for about a week.

That said, I think it should be fairly straightforward to address the reviewers' comments. There was substantial criticism of the writing, but that should be easy for you to correct. I really like to be consistent with tenses, and my feeling is that anything the authors of a paper did for that study should be presented in past tense and any established background material (i.e., material from the literature) should be present tense (unless the material from the literature refers specifically to something that happened in the past). Please adhere to that as you write your revision.

Reviewer 3 was the most critical of the scientific content. However, I recommend a minor revision because I do not feel it is necessary to conduct reanalyses of published data as s/he recommended. Obviously, a component of any study is convincing the audience that your methodologies are useful, but I question the practicality of doing a full-scale reanalysis of prior information. However, do take care to make sure your background is clear, even to less specialized readers.

·

Basic reporting

Text is fairly clear and unambiguous. Some additional literature citations could be added which I point out. The paper is very professionally organized, data are shared, and it is self contained - no comments there.

Experimental design

No comments

Validity of the findings

No comments.

Additional comments

This paper contains two parts, 1) a commentary and suggested revisions on how to score primary molt in birds and 2) a presentation of molt in Gray-winged Trumpeter (Psophia crepitans) [I suggest authors use common name somewhere, if only in parentheses when initially referred to] Both sections are presented fairly well and are interesting and worthy of presentation. Understanding remegial molt sequences in birds is often as much of an art as a science, and I like to look at wings and deduce what is going on in more of a holistic manner. But I also agree that molt summary tables are also helpful or necessary to figure things out, especially in very complicated cases such as cuckoos.

My concerns are summarized by specific comments throughout the text.

Abstract, lines 27-31: The subject of this sentence is stated to be primary replacement but (iv) expands it to the entire wing. This should be restated somehow. Furthermore, (ii) and (iii) are largely if not entirely overlapping and I suggest they be combined in this sentence.

Abstract: Add sentence at end on the evolution of molts in Gruiformes (see below).

Lines 73-77: Run-on sentence - break it into 2 or 3.

Line 80 - Howell (2000) not in lit cited. Maybe Howell (2010) or Heermanns (2000) was meant. In either case, there are better primary sources in this case: I believe that Dorward (1962) was the first to fully document this pattern in Staffelmauser (stepwise) molt and Stresemann and Stresemann (S&S 1966) and several others cited in this paper could be included instead.

Dorward, D.F. 1962. Comparative biology of the White Booby and the Brown Booby Sula spp. at Ascension Island. Ibis 103b:174-220.

Line 82 (and several other places in this ms.): I'm not convinced that a series in the primaries should be linked with a series in the secondaries (vis-a-vis a continuous wave from p1 to s1 or vice versa) and would recommend that this concept be dropped in this paper (it is somewhat peripheral anyway as sequence of secondary replacement is not treated in depth). In specimens I've examined of albatrosses and limpkins, molt of secondaries begins when series among primaries are mid-stream and not when the primary molt reaches p1. I realize that Edwards and Rohwer (2005) suggested this but I believe their data could just as easily be explained as a terminus at p1 and a node at s1 - just coincidentally (and unusually among birds) that the two series are both proximal. A node at s1 seems so evolutionarily fixed in almost all birds that I would favor this explanation in albatross and Psophiidae as well.

Lines 104-116 - Some of this is not relevant to molt and I would suggest reducing this paragraph by 2-3 sentences. On the other hand, it would be good to add here that molt has been documented as proximal by S&S (1966). Note also the short chapter on molt in Psophiidae in Johnson and Wolfe (2018, Studies in Avian Biology 51). Here they mistakenly cite S&S as molt being distal (undoubtedly due to use of the confusing term "ascendant sequence" by them and others in the European literature). They also cite Sherman (1995) on molt (Auk - available through SORA) but not much is mentioned here other than that juveniles retain brown plumage until a year of age, at which time they molt and become indistinguishable from adults. Could be worth citing though. Finally, the sentence on lines 115-116 seems incongruous to me - Because... primaries fade little in the forest.... wear was used for scoring... At one time I also looked at molt in Psophiidae specimens and concluded possible Staffelmauser (as mentioned in J&W 2018) but was unsure, probably because I was assuming distal replacement and may have been confused about what I was looking at.

Line 28 and elsewhere: Most use "juvenile" as a term to indicate birds prior to their first molt. It would seem better to replace this with "first-year" so as to include those undergoing molt, or specifically refer to these molting birds as undergoing the second prebasic molt.

Line 170 and elsewhere: I would also say "suspend" in addition to "arrest" here as suspension of molt (e.g. in raptors during breeding or migration; Pyle 2005) can also occur and confound interpretation of molt sequences in the same manner as arrested molts can.

Line 175: replace "before" with "during". The replacement occurs during incubation, then molt suspends for chick feeding and resumes after chicks fledge. Pyle (2005) can be added as a citation here.

Line 181 (and elsewhere?): "flight-feather" should be hyphenated is such usages.

Line 183: add "P" before "10"

Lines 220-222: I have studied secondary replacement in birds that undergo Staffelmauser (e.g., Pyle 2005 - cited, 2006, 2008; 2013 - cited). In (most) diastatatxic species it works the same as with primaries with waves beginning proximally at s1 and s5 and distally from the tertials and then with new waves commencing before feather termini (at s4 and somewhere between s6 and the tertials) are reached. Interestingly, Bostwick and Brady (1992 - Auk) report that Psophiidae are eutaxic and if correct these species might indeed undergo some other replacement strategy among secondaries. This could be mentioned in connection with secondary replacement in Psophiidae.

Pyle, P. 2006. Staffelmauser and other adaptive wing-molt strategies in larger birds. Western Birds 37:179-185.
http://www.birdpop.org/docs/pubs/Pyle_2006_Staffelmauser_and_Other_Adaptive_Strategies_in_Larger_Birds.pdf

Pyle, P. 2008. Identification Guide to North American Birds. Part 2. Slate Creek Press, Point Reyes Station, CA.

Lines 233-235: I am familiar with the Rohwer-Howell dispute on molt in albatrosses and have not taken sides. But in respect to Howell, the dispute is not over how many series albatross have, just where the nodes and termini are. In any case, I would suggest deleting this here as it seems tangential and a bit of a petty dig.

Lines 241-256: This paragraph seems out of place and a bit tangential and could be deleted or shortened and moved to the Discussion. As an aside (line 242 and elsewhere), I have had success interpreting molt sequences among secondaries on specimens (e.g., Pyle 2013 - cited) and although more difficult than primaries it is not impossible. Futhermore...

Line 262:... it can be done without damage to specimens so I would delete this inference here.

Line 272 (and elsewhere): The term "transient node" seemed fine to me at first but then I started wondering if it might add more confusion than help to the lexicon. As used here it has little evolutionary or even practical value. Maybe "false node" would be a bit clearer, but I might also consider resisting the definition of "transient node".

Line 283: I've also noticed this problem in birds with short p10s and it's great to have this pointed out in writing.

Lines 299-305: This paragraph caused me to consult the Supplemental Excel file and I have a couple of suggestions to this: Most molt-scoring tables have p1 to the left and p10 to the right and for comparative purposes it might be worth switching this order. As I have used and studied these tables, I became confused looking at this one because of this. Second, in the heading please put "P" before each primary number as this will be helpful. Third, repeat the heading above the juveniles so we know what primaries we are looking at. Regarding this paragraph, see above suggestion on age terminology. I would only consider AMNH 178924 to possibly be a "juvenile" depending on what's going on with p1 (see below). The reference to AMNH 431790 requires consultation of the Excel file to understand so I would suggest a bit more explanation here. I agree with authors' ultimate reasoning and it might be pointed out somewhere in the paper (if not already) that the p10 and p3 waves would appear to commence simultaneously (based on AMNH 125285 and a couple of the adult specimens) and thus the inner wave should probably complete at about the same time as p7 or p8 are being replaced in the outer wave.

Lines 306-315: I'm having a lot of trouble with this paragraph and suggest it be deleted or recast. First, see above regarding my suspicions about a correlation between secondary and primary waves. Second, parrots and falcons should not be cited here as in these cases the waves are proximal in the inner primaries and distal in the outer secondaries and so both p1 and s1 should be regarded as termini. Quite different from assuming a wave among one tract carries over to the other tract. Third, since secondaries were not looked at much in this paper how can it be "suspected" that they are "likely" connected? This assumption would entail the wave in the secondaries being distal, rare at best in birds (parrots, falcons, and Galliformes are all I know of - maybe Limpkin - which would be of interest, but this needs confirmation). Fourth, looking at the Excel file regarding these six birds (and the juvenile 178924) I don't see why another interpretation couldn't be that Psophiidae have another distal node at p1? Since this is a common occurrence in birds, perhaps it occurs here as well and there is a terminus at p2 (cf 272991 that would appear to be completing molt with p4 and p2 - also 257106 with p3 and p1 new and p2 old). See also 283174 which has p10 and p1 dropped at same time. Or perhaps more likely is that Staffelmauser patterns are occurring among p1-p3 as well - which could also help explain apparent nodes at p2 (125279, 178923). I would like to see all of these possibilities discussed more thoroughly along with or in lieu of a cross s1-p1 wave.

Lines 328-331: Could cite Pyle 2005, 2006, 2008 here as well.

Lines 338-342: recast or restate this as well (see above). While two waves would be hard or impossible to document among three primaries (though what about 283174?), I believe that some of the evidence still suggests suspended or arrested molts among these feathers (whatever the nodal arrangement). In this case a more holistic look at it (e.g., exact degrees of wear among the three primaries and in context to the precise scenario in the outer primaries) could help us interpret this.

Lines 351-364: This paragraph is also rather speculative and tangential and may not need to be included. Collection biases are well known for all sorts of reasons and could also explain the low numbers of younger birds. Lines 353-354 - this should have been easy to infer by looking at specimens to see if any had mixed rectrices but were not in active molt, comparing rectrix patterns to remex patterns, considering wear to juvenile rectrices, etc. I would drop this. Lines 362-363 - Johnson and Wolfe (2018) conclude that rectrices are not replaced until the second prebasic molt and I would agree with this.

Line 378: Add Pyle (2013) - falcons and parrots.

Line 387: Recast as suggested above.

Lines 393-412: It would be great to include some thoughts on the evolution of molts in Gruiformes here. First, I believe that rails may also molt primaries proximally. I saw this cited (as "ascendant") in S&S (1966) with reference to an older German paper(I can't immediately locate my copy of S&S or I would check the reference). I believe it is also mentioned in Taylor (2010 - Rails of the World). Even though most rails are reported to molt remiges "simultaneously" I think they may just molt them rapidly but in a sequence, as Thompson suggested for puffins (and I've seen in photos and specimens of various alcids as well). In addition to the need for molt scoring tables for Limpkins I would include cranes since their remigial molt sequences are also complex and poorly understood. Molts in finfoots and sungrebes could also be discussed, if there is any information out there. A finding of proximal replacement of primaries in most or all Gruiformes would suggest that this is an evolutionarily fixed trait as suggested for falcons and parrots.

Lines 415-416: Would albatrosses also fit this description?

Hope this helps

Peter Pyle
[email protected]

Reviewer 2 ·

Basic reporting

I dislike spending time editing grammatical errors and clarity during reviews. Thus, I only edited the abstract for flow and grammar, and encourage the authors to carefully evaluate the entire manuscript in a similar fashion. See below.

General comment: overuse of “the” throughout. For example, “…Using the 80 specimens of Psophia crepitans at the American Museum of Natural History…” is rendered clear without *the*: “…Using 80 specimens of Psophia crepitans at the American Museum of Natural History …”

Given the authors renowned publications, I was a little surprised at the choppiness and lack-of-clarity in the abstract and introduction. For example, in the statement above, why use a present participle in the past tense? This is sloppy in my opinion. It is much clearer with the past tense: We used 80 specimens of…


ABSTRACT

Page 2 line 26 “…the first illustration for generating molt summary tables…” is awkward for two reasons. First, most readers will be unfamiliar with ‘molt summary tables’; thus, describing molt summary tables using the ambiguous term “illustration” may be interpreted as a noun – a literal illustration. Did people draw these tables? Second, try to avoid confusing present participles (words ending in “ing” such as “generating”) in the past tense. Many scientific style guides suggest limiting present participles in general.

Page 2 line 26-27 “…molt summary tables used a temperate passerine…” is awkward. You switch from plural (tables) to singular (a temperate passerine). To the uniformed reader this comes off as contradictory.

Page 2 line 27-28 The following statement makes little sense: “But primary replacement in temperate passerines is far too simple to address the complexities of generating summary tables for…” taken literally, it comes off as passerines are at fault for our collective inability to create complex molt summary tables. Let me be clear, I know what you mean. I suggest clarifying the sentence to express your sentiment.

Page 2 line 36 “…reverse that of…” is awkward.

Page line 38-39 “we develop molt tables that show that the sequence of primary replacement is, indeed, proximal.” Think critically about sentence structure. This is sloppy. “We develop” is present tense when it should be past tense. “…that show that the…” is awkward. The sentence can easily be improved (i.e. …our molt tables found that Psophia crepitans undergoes a proximal sequence of primary replacement).
Please follow my brief example and critically review the writing prior to resubmission.

TABLES

Need to define rows and columns. The reader is left to intuit the details of this table. What is #p and #d and why are they in their own rows without headers? Strange and incomplete presentation of a table.

Experimental design

You have placed the onus on the reader to seek out definitions of molt summary tables. Specifically, the majority of the reading audience is likely unfamiliar with molt summary tables and how they differ from similar data derived from “snap shots”. You need a brief, yet thorough definition of a molt summary table prior to discussing nuances of the system. This is particularly important for the results section where the reader is presumed to be familiar with molt table protocols, once again placing the onus on the reader to familiarize themselves with protocols before interpreting the results.

Validity of the findings

RESULTS

I have always battled with differences between suspended and arrested molts, and how to differentiate them. The authors do a nice job detailing potential difficulties in identifying arrested molts and transient nodes when remiges don’t accrue much wear and I suggest expanding the definitions to include suspended molts. In general, it would be good to define some common terms, in addition to arrested and suspended, specific definitions of “molt series” would be helpful.

In the Primary replacement in Psophia crepitans section, please remind readers how many primaries trumpeters have prior to discussing the nuances of their replacement. Additionally, I suggest using terminology that would provide some insight into putative homology. For example, for the juvenile trumpeters that were found replacing primaries: was this molt a post-juvenile molt occurring shortly after acquiring the juvenile plumage? Or is this a molt which occurs much later in life, and would be similar (putatively homologous) with an adult (definitive) prebasic (post-breeding) molt? This information would be value added for our broader understanding of tropical bird molt.

Reviewer 3 ·

Basic reporting

Authors use specialized terms for descriptions of molt systems but do not define them clearly in the earlier sections of the manuscript (they should be defined in Introduction or at the beginning of Methods preferably with good illustrations). Defining the terms and illustrating them will make their descriptions of molt summary tables much easier to understand.

In the tables and supplemental information (raw data), some abbreviations need to be explained or spelled out (e.g., # p, # d, pp, p, d).

I found some of the result sections difficult to follow and believe that they need to to revised to clarify or make them flow better (see attached PDF).

Experimental design

The descriptions of the new molt summary table are very difficult to follow as some statements seem to be contradictory (e.g., assessment of directionality). The new method is supposed to simplify the previous methods, but there is no clear descriptions of the steps to follow to construct new tables.

Validity of the findings

I am not yet convinced that the new method improves the understanding of molt series. It will be more convincing if the authors reanalyze previously published data and show that it clarifies the results or leads to different conclusions.

There is no evidence to show that P1 is part of the P1–3 molt series (as for 6 of 9 observations, it was scored as nodal).

Discussion on the very low rate of (re)production is highly speculative and doesn't seem to belong to this manuscript.

Additional comments

Please refer to the attached PDF for more detailed comments.

Annotated reviews are not available for download in order to protect the identity of reviewers who chose to remain anonymous.

---

## Round 0.2 · Minor Revisions

I apologize for getting this back to you a bit late. Between travel and feeling a bit sick I just dropped the ball. I feel bad about that and do feel that I owe you this apology.

Regardless, I am happy with the manuscript but both reviewers did have some editorial comments. They all seem minor and very easy to deal with so I felt I should ask for you to comb through and deal with them. But in terms of comments we are all happy and if you make them I consider this accepted for publication (I normally don't say this until the last email -- but the reviewers' comments are so trivial and based on english that I'm going to go ahead and say it). Hopefully you can get it back to me shortly - I'm looking forward to it!

·

Basic reporting

Fine (no comment)

Experimental design

Fine (no comment)

Validity of the findings

Fine (no comment)

Additional comments

Review of Rohwer and Rohwer, Primary molt in Psophia and simpler molt summary tables, for PeerJ

Comments to authors

This paper contains two parts, 1) a commentary and suggested revisions on how to score primary molt in birds and 2) a presentation of molt in Gray-winged Trumpeter (Psophia crepitans) [I suggest authors use common name somewhere, if only in parentheses when initially referred to] Both sections are presented fairly well and are interesting and worthy of presentation. Understanding remigial molt sequences in birds is often as much of an art as a science, and I like to look at wings and deduce what is going on in more of a holistic manner. But I also agree that molt summary tables are helpful or necessary to figure things out, especially in very complicated cases such as cuckoos.

My concerns are summarized by specific comments throughout the text.

Abstract, lines 27-31: The subject of this sentence is stated to be primary replacement but (iv) expands it to the entire wing. This should be restated somehow. Furthermore, (ii) and (iii) are largely if not entirely overlapping and I suggest they be combined in this sentence.

Abstract: Add sentence at end on the evolution of molts in Gruiformes (see below).

Lines 73-77: Run-on sentence - break it into 2 or 3.

Line 80 - Howell (2000) not in lit cited. Maybe Howell (2010) or Heermanns (2000) was meant. In either case, there are better primary sources in this case: I believe that Dorward (1962) was the first to fully document this pattern in Staffelmauser (stepwise) molt and Stresemann and Stresemann (S&S 1966) and several others cited in this paper could be included instead.

Dorward, D.F. 1962. Comparative biology of the White Booby and the Brown Booby Sula spp. at Ascension Island. Ibis 103b:174-220.

Line 82 (and several other places in this ms.): I'm not convinced that a series in the primaries should be linked with a series in the secondaries (vis-a-vis a continuous wave from p1 to s1 or vice versa) and would recommend that this concept be dropped in this paper (it is somewhat peripheral anyway as sequence of secondary replacement is not treated in depth). In specimens I've examined of albatrosses and limpkins, molt of secondaries begins when series among primaries are mid-stream and not when the primary molt reaches p1. I realize that Edwards and Rohwer (2005) suggested this but I believe their data could just as easily be explained as a terminus at p1 and a node at s1 - just coincidentally (and unusually among birds) that the two series are both proximal. A node at s1 seems so evolutionarily fixed in almost all birds that I would favor this explanation in albatross and Psophiidae as well.

Lines 104-116 - Some of this is not relevant to molt and I would suggest reducing this paragraph by 2-3 sentences. On the other hand, it would be good to add here that molt has been documented as proximal by S&S (1966). Note also the short chapter on molt in Psophiidae in Johnson and Wolfe (2018, Studies in Avian Biology 51). Here they mistakenly cite S&S as molt being distal (undoubtedly due to use of the confusing term "ascendant sequence" by them and others in the European literature). They also cite Sherman (1995) on molt (Auk - available through SORA) but not much is mentioned here other than that juveniles retain brown plumage until a year of age, at which time they molt and become indistinguishable from adults. Could be worth citing though. Finally, the sentence on lines 115-116 seems incongruous to me - Because... primaries fade little in the forest.... wear was used for scoring... At one time I also looked at molt in Psophiidae specimens and concluded possible Staffelmauser (as mentioned in J&W 2018) but was unsure, probably because I was assuming distal replacement and may have been confused about what I was looking at.

Line 28 and elsewhere: Most use "juvenile" as a term to indicate birds prior to their first molt. It would seem better to replace this with "first-year" so as to include those undergoing molt, or specifically refer to these molting birds as undergoing the second prebasic molt.

Line 170 and elsewhere: I would also say "suspend" in addition to "arrest" here as suspension of molt (e.g. in raptors during breeding or migration; Pyle 2005) can also occur and confound interpretation of molt sequences in the same manner as arrested molts can.

Line 175: replace "before" with "during". The replacement occurs during incubation, then molt suspends for chick feeding and resumes after chicks fledge. Pyle (2005) can be added as a citation here.

Line 181 (and elsewhere?): "flight-feather" should be hyphenated is such usages.

Line 183: add "P" before "10"

Lines 220-222: I have studied secondary replacement in birds that undergo Staffelmauser (e.g., Pyle 2005 - cited, 2006, 2008; 2013 - cited). In (most) diastatatxic species it works the same as with primaries with waves beginning proximally at s1 and s5 and distally from the tertials and then with new waves commencing before feather termini (at s4 and somewhere between s6 and the tertials) are reached. Interestingly, Bostwick and Brady (1992 - Auk) report that Psophiidae are eutaxic and if correct these species might indeed undergo some other replacement strategy among secondaries. This could be mentioned in connection with secondary replacement in Psophiidae.

Pyle, P. 2006. Staffelmauser and other adaptive wing-molt strategies in larger birds. Western Birds 37:179-185.
http://www.birdpop.org/docs/pubs/Pyle_2006_Staffelmauser_and_Other_Adaptive_Strategies_in_Larger_Birds.pdf

Pyle, P. 2008. Identification Guide to North American Birds. Part 2. Slate Creek Press, Point Reyes Station, CA.

Lines 233-235: I am familiar with the Rohwer-Howell dispute on molt in albatrosses and have not taken sides. But in respect to Howell, the dispute is not over how many series albatross have, just where the nodes and termini are. In any case, I would suggest deleting this here as it seems tangential and a bit of a petty dig.

Lines 241-256: This paragraph seems out of place and a bit tangential and could be deleted or shortened and moved to the Discussion. As an aside (line 242 and elsewhere), I have had success interpreting molt sequences among secondaries on specimens (e.g., Pyle 2013 - cited) and although more difficult than primaries it is not impossible. Futhermore...

Line 262:... it can be done without damage to specimens so I would delete this inference here.

Line 272 (and elsewhere): The term "transient node" seemed fine to me at first but then I started wondering if it might add more confusion than help to the lexicon. As used here it has little evolutionary or even practical value. Maybe "false node" would be a bit clearer, but I might also consider resisting the definition of "transient node".

Line 283: I've also noticed this problem in birds with short p10s and it's great to have this pointed out in writing.

Lines 299-305: This paragraph caused me to consult the Supplemental Excel file and I have a couple of suggestions to this: Most molt-scoring tables have p1 to the left and p10 to the right and for comparative purposes it might be worth switching this order. As I have used and studied these tables, I became confused looking at this one because of this. Second, in the heading please put "P" before each primary number as this will be helpful. Third, repeat the heading above the juveniles so we know what primaries we are looking at. Regarding this paragraph, see above suggestion on age terminology. I would only consider AMNH 178924 to possibly be a "juvenile" depending on what's going on with p1 (see below). The reference to AMNH 431790 requires consultation of the Excel file to understand so I would suggest a bit more explanation here. I agree with authors' ultimate reasoning and it might be pointed out somewhere in the paper (if not already) that the p10 and p3 waves would appear to commence simultaneously (based on AMNH 125285 and a couple of the adult specimens) and thus the inner wave should probably complete at about the same time as p7 or p8 are being replaced in the outer wave.

Lines 306-315: I'm having a lot of trouble with this paragraph and suggest it be deleted or recast. First, see above regarding my suspicions about a correlation between secondary and primary waves. Second, parrots and falcons should not be cited here as in these cases the waves are proximal in the inner primaries and distal in the outer secondaries and so both p1 and s1 should be regarded as termini. Quite different from assuming a wave among one tract carries over to the other tract. Third, since secondaries were not looked at much in this paper how can it be "suspected" that they are "likely" connected? This assumption would entail the wave in the secondaries being distal, rare at best in birds (parrots, falcons, and Galliformes are all I know of - maybe Limpkin - which would be of interest, but this needs confirmation). Fourth, looking at the Excel file regarding these six birds (and the juvenile 178924) I don't see why another interpretation couldn't be that Psophiidae have another distal node at p1? Since this is a common occurrence in birds, perhaps it occurs here as well and there is a terminus at p2 (cf 272991 that would appear to be completing molt with p4 and p2 - also 257106 with p3 and p1 new and p2 old). See also 283174 which has p10 and p1 dropped at same time. Or perhaps more likely is that Staffelmauser patterns are occurring among p1-p3 as well - which could also help explain apparent nodes at p2 (125279, 178923). I would like to see all of these possibilities discussed more thoroughly along with or in lieu of a cross s1-p1 wave.

Lines 328-331: Could cite Pyle 2005, 2006, 2008 here as well.

Lines 338-342: recast or restate this as well (see above). While two waves would be hard or impossible to document among three primaries (though what about 283174?), I believe that some of the evidence still suggests suspended or arrested molts among these feathers (whatever the nodal arrangement). In this case a more holistic look at it (e.g., exact degrees of wear among the three primaries and in context to the precise scenario in the outer primaries) could help us interpret this.

Lines 351-364: This paragraph is also rather speculative and tangential and may not need to be included. Collection biases are well known for all sorts of reasons and could also explain the low numbers of younger birds. Lines 353-354 - this should have been easy to infer by looking at specimens to see if any had mixed rectrices but were not in active molt, comparing rectrix patterns to remex patterns, considering wear to juvenile rectrices, etc. I would drop this. Lines 362-363 - Johnson and Wolfe (2018) conclude that rectrices are not replaced until the second prebasic molt and I would agree with this.

Line 378: Add Pyle (2013) - falcons and parrots.

Line 387: Recast as suggested above.

Lines 393-412: It would be great to include some thoughts on the evolution of molts in Gruiformes here. First, I believe that rails may also molt primaries proximally. I saw this cited (as "ascendant") in S&S (1966) with reference to an older German paper(I can't immediately locate my copy of S&S or I would check the reference). I believe it is also mentioned in Taylor (2010 - Rails of the World). Even though most rails are reported to molt remiges "simultaneously" I think they may just molt them rapidly but in a sequence, as Thompson suggested for puffins (and I've seen in photos and specimens of various alcids as well). In addition to the need for molt scoring tables for Limpkins I would include cranes since their remigial molt sequences are also complex and poorly understood. Molts in finfoots and sungrebes could also be discussed, if there is any information out there. A finding of proximal replacement of primaries in most or all Gruiformes would suggest that this is an evolutionarily fixed trait as suggested for falcons and parrots.

Lines 415-416: Would albatrosses also fit this description?

Hope this helps

Peter Pyle
[email protected]


Comments to Editor:

I am not yet familiar with PeerJ's style but suggest a couple of the paragraphs and subjects covered by this paper are not needed here - either too speculative or too tangential. It will be obvious from my comments to authors which these are. Editor can help decide how to instruct authors on these.

Review of Revision

I'm pleased to see how much work was put into the revision. The added material on molt in Gruiforms greatly increases the value of the paper. Other great additions include much improved descriptions of how molt tables work, excellent summaries of the interpretation, meaning, and value of nodal and terminal feathers, and the new section pointing out confusion of the terms 'ascendent' and 'descendent' (finally in print!). I'm also satisfied with almost all of the smaller fixes attended to.This is a very clearly worded and valuable contribution to the study of primary molt, of several aspects.

However, I continue to have trouble with the emphasis of a primary-secondary (p1-s1) crossover here. It seems incongruous to strongly emphasize problems of previous molt tables for interpretation of species with multiple series and staffelmauser (e.g., Abstract lines 26-30, Introduction lines 135-138, and Discussion lines 509-512) while at the same time having complete faith in the results of a 2005 paper on species that have both of these problems! I appreciate the insertion of "apparently" twice while referring to this crossover (lines 86, 207) but I still find the wording assuming this crossover to be dangerously conclusive. Mind you I'm not arguing that this crossover doesn't necessarily exist, just that I don't think Edwards and Rohwer (2005) conclusively showed this with the molt tables published there (though much else in that paper is of value). As I recall I reviewed that paper and pointed this out then, but my cautions were not carried over to print. It may be a place where we agree to disagree (not on the fact but on the confidence in the fact) but I continue to recommend this idea be further softened or reduced here, especially since it is a bit peripheral to all of the other great work in this ms. on primary molt series in Gruiformes.

Specific comments

Abstract, line 36: Replace "P10" with "the outer primary" since P10 is not defined yet and given different American vs. European numerical nomenclatures as pointed out elsewhere in the paper.

Line 160: I would replace "two" with "two or more". I have data suggesting that termini in large birds with many secondaries can be among up to 4 feathers between s5 and the tertials. Check also Edwards and Rohwer (2005) on this.

Lines 174-194: Somewhere it needs to be emphasized that the first replacement of juvenile feathers, occurring at the 2nd, 2nd-3rd, or 2nd-4th prebasic molts in staffelmauser species, is the best way to determine stable vs. transient nodes. To me the specimens undergoing this (or these) first molts are most valuable, as this first replacement sequence appears to be fixed with respect to later replacement waves.

Lines 197-199: Pyle (2008 pp 23-25) covers this topic in detail, the result of examination of 100's or 1000's of specimens treated in that work (and see also Pyle 2006). Although I did not create molt tables for every species (for obvious reasons) I believe it deserves a mention here alongside Edwards and Rohwer, as I am convinced the conclusions are at least largely correct.

Lines 214-216 and 248-250 are redundant.

Line 263: Replace "rectrices of immature birds" with "juvenile rectrices" and (later in line) "those" with "basic feathers"

Lines 313-315: In addition to my overall concerns about this (above) - I'm not sure I follow the logic here. It seems P1 would be nodal whether or not there is a crossover to S1.

Line 333 replace "hatch-year" with "first-year" to be consistent with earlier usage and because most inferred replacement occurs in the second calendar year rather than the hatching year.

Line 375: Insert "Gruiform" after "new world"

Line 406: clarify - there may be a typo

Line 529: Since cranes share several similarities to molt in other Gruiforms I think it OK to speculate that they will also be found to have proximal replacement. Amazing this hasn't been documented yet!

Lines 527-529 and 540-541 are redundant.

Line 583: Not sure if "Ironically" is appropriate here, especially if Stresemanns earlier suspected it. After this, replace "molt data" with "molt-sequence data" as sequence seems to be the only part of overall molt strategies that shows synapomorphy (Pyle 2013). Sibley and Ahlquist (1990) also discuss this quite a bit and you might want to reference them. Finally, a sentence pointing out that proximal replacement likely evolved in an ancestral Gruiform (before the families split) would certainly be worth including in this section, along with how fixed it has remained, with the possible exception of the p3 node in trumpeters being derived.

Lines 585-587: Again it seems to me that albatross could combine multiple series and staffaulmauser, but parrots definitely do, as documented by Pyle (2013).

Hope this helps,

Peter

Reviewer 3 ·

Basic reporting

I was delighted to see that the authors took the reviewers’ comments seriously and made the manuscript significantly better . The manuscript is well organized and very easy to read, and all the important concepts are well described/explained. Both Introduction and Results are much more accessible to a general audience, and detailed background in life histories and previous molt studies of Gruiform birds is particularly informative for understanding the importance of this study.

Experimental design

The authors explained quite well how their study fills knowledge gaps. The methods are much better explained.

Validity of the findings

Added data from multiple members of the Gruiformes made this study more compelling and robust. They proved that their methods and findings can be more generalized and widely applied.

Additional comments

Here are some minor specific comments:

Lines 75–76: The order of the two references should be reversed.

Lines 199–204: I am confused about the difference between rate and frequency of feather replacement. Does the rate mean how fast the feathers are replaced? How does it help identify stable nodes?

Line 371: Should the ending be “.”?

Lines 410–415: Could this bias also be a result of bias in collecting dates (i.e., seasonal bias in the state of molt in the specimens)?

Figure 1: The Y axis labels should be in percentages (fractions of 100: e.g., 0, 20, 40, 60, 80).

---

## Round 0.3 · accepted · Accept

Thank you so much for the revisions. I have carefully reviewed the manuscript and I am happy to accept the manuscript. I look forward to seeing it in print!

#